# Dual-gated single-molecule field-effect transistors beyond Moore's law

Linan Meng[1,2,7], Na Xin[1,7], Chen Hu[3,7], Hassan Al Sabea[4,7], Miao Zhang[5,7], Hongyu Jiang[2,6], Yiru Ji[2,6], Chuancheng Jia[5], Zhuang Yan [1], Qinghua Zhang[2], Lin Gu [2], Xiaoyan He[4], Pramila Selvanathan[4], Lucie Norel[4], Stéphane Rigaut [4✉], Hong Guo[3✉], Sheng Meng [2,6✉] & Xuefeng Guo [1,5✉]

As conventional silicon-based transistors are fast approaching the physical limit, it is essential to seek alternative candidates, which should be compatible with or even replace microelectronics in the future. Here, we report a robust solid-state single-molecule field-effect transistor architecture using graphene source/drain electrodes and a metal back-gate electrode. The transistor is constructed by a single dinuclear ruthenium-diarylethene (Ru-DAE) complex, acting as the conducting channel, connecting covalently with nanogapped graphene electrodes, providing field-effect behaviors with a maximum on/off ratio exceeding three orders of magnitude. Use of ultrathin high-$k$ metal oxides as the dielectric layers is key in successfully achieving such a high performance. Additionally, Ru-DAE preserves its intrinsic photoisomerisation property, which enables a reversible photoswitching function. Both experimental and theoretical results demonstrate these distinct dual-gated behaviors consistently at the single-molecule level, which helps to develop the different technology for creation of practical ultraminiaturised functional electrical circuits beyond Moore's law.

[1] Beijing National Laboratory for Molecular Sciences, National Biomedical Imaging Center, College of Chemistry and Molecular Engineering, Peking University, 292 Chengfu Road, Haidian District, Beijing 100871, PR China. [2] Beijing National Laboratory for Condensed Matter Physics, Institute of Physics, Chinese Academy of Sciences, Beijing 100190, PR China. [3] Center for the Physics of Materials and Department of Physics, McGill University, Montreal, QC H3A 2T8, Canada. [4] Univ Rennes, CNRS, ISCR (Institut des Sciences Chimiques de Rennes)-UMR 6226, F-35000 Rennes, France. [5] Center of Single-Molecule Sciences, Institute of Modern Optics, Frontiers Science Center for New Organic Matter, College of Electronic Information and Optical Engineering, Nankai University, 38 Tongyan Road, Jinnan District, Tianjin 300350, PR China. [6] University of Chinese Academy of Sciences, Beijing 100049, PR China. [7] These authors contributed equally: Linan Meng, Na Xin, Chen Hu, Hassan Al Sabea, Miao Zhang. ✉email: stephane.rigaut@univ-rennes1.fr; hong.guo@mcgill.ca; smeng@iphy.ac.cn; guoxf@pku.edu.cn

In 2016, in light of the increasingly urgent need to continue scaling down silicon-based electronic devices, the worldwide semiconductor industry formally acknowledged that Moore's law was nearing its end and announced the so-called "More than Moore strategy", which focuses on both miniaturization and functionality increase[1]. Single-molecule electronics, which has the ultimate goal of electronic device miniaturization through use of nanometer-scale active components, can provide new insights that will help the industry to meet the requirements of this strategy[2–6]. To this end, discrete approaches have been developed to fabricate electronic devices based on individual molecules[7,8]. An important step to bring single-molecule electronics as a complementary technology for microelectronic devices in this process would be the fabrication of a reliable unimolecular field-effect transistor (FET)[9]. Indeed, as we cannot construct an electrical circuit without FETs, which are the basic building blocks of current computer circuitry, this achievement will be the ultimate test to evaluate the usefulness of these devices in conventional integrated circuit architectures. In this work, we demonstrate dual-gated single-molecule transistors in a solid device configuration, where the high on/off ratio of single-molecule FETs is achieved.

We previously demonstrated that molecules can be used as a pivotal element to achieve reproducible photo-controlled conductance switching with high accuracy[10]. To move forward, we now aim to construct practical single-molecule FETs that can address a double challenge with (1) the achievement of an elevated on/off ratio suitable for real applications and (2) the remote control of this performance to obtain a high level of complexity for advanced functional applications. Indeed, the single-molecule transistors reported to date have shown limited potential for use in real applications because of their low on/off ratios (generally less than 10) or their solution-based operating conditions[11–16]. Here, we demonstrate progress in constructing a stable solid-state single-molecule FET with an exceptional on/off ratio that consists of a single dinuclear ruthenium-diarylethene (Ru-DAE) complex connected between nanogapped graphene electrodes with high-$k$ HfO$_2$/Al$_2$O$_3$ acting as the dielectric layer.

As illustrated in Fig. 1a, individual dinuclear Ru-DAE molecules containing two Ru fragments [H$_2$N–C$_6$H$_4$–C≡C(dppe)$_2$Ru]$^+$ (dppe = 1,2-bis(diphenylphosphino)ethane) at the mirror-symmetric positions of the two thiophene rings were connected to the source and drain graphene electrodes (Details of molecular synthesis/characterization and junction structure are provided in the Methods Section and Supplementary Figs. 1−4) and combined with a bottom-gate control electrode to form a practical FET configuration. The use of graphene as the source/drain electrode material offers more than one merit. Firstly, it avoids the gate screening that can occur in metal-molecule-metal junctions because graphene has atomic thickness and offers good compatibility and robust covalent binding with single molecules. Secondly, the good stability of both the graphene electrode materials and the covalent amide linkages make the contacts of the resulting fabricated junctions more robust[17–19] than the metal-sulfur bonds (e.g., Au–S bonds) used in other molecular junctions, thus ensuring a reliable (i.e., stable and reproducible) exploration of the device function from the perspective of the intrinsic properties of the connected molecule.

One of the most formidable challenges to be faced when miniaturizing these devices to the molecular level is the short channel effect, which hampers practical fabrication of high-performance single-molecule FETs[6,20–22]. An efficient way to overcome this issue is to use ultrathin high-$k$ dielectric materials to enhance the gate coupling and thus enhance the overall device performance. For this reason, we selected thermally evaporated Al metal as the gate electrode material, which was covered with naturally oxidized Al$_2$O$_3$[23], together with HfO$_2$ prepared using the sol-gel method as the dielectric material[6,24]. The optical and cross-sectional scanning transmission electron microscope images of the representative FET device shown in Fig. 1b, c and in Supplementary Fig. 5 illustrate the perfect atomic flatness of the dense dielectric bilayer (thickness: ~10 nm), which thus promises efficient gate modulation (Supplementary Fig. 6) and negligible leakage currents, as will be

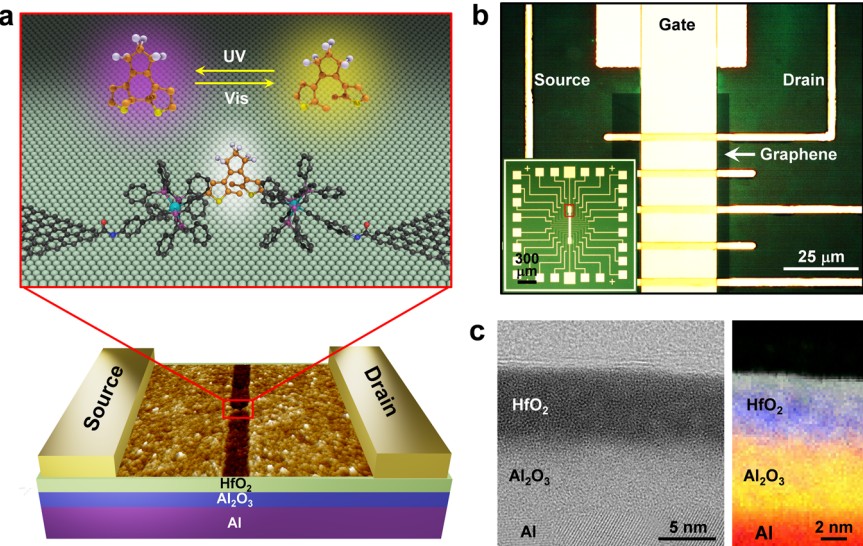

**Fig. 1 Device diagrams of a graphene-Ru-DAE-graphene single-molecule FET. a** Schematic representation of the device structure. Bottom: Atomic force microscopic image of nanogapped graphene point contacts with the bottom gate. Top: Schematic of the device center that highlights the reversible isomerisation of the DAE unit between ring-open and ring-closed forms that are triggered by optical stimuli. **b** Optical images of a graphene-Ru-DAE-graphene single-molecule FET array with a common bottom gate based on a HfO$_2$/Al$_2$O$_3$/Al multilayer. The inset shows the complete pattern, where the central region marked by a red circle is enlarged for clarity. **c** Left: Cross-sectional scanning transmission electron microscope (STEM) image of the HfO$_2$/Al$_2$O$_3$/Al multilayer structure. The sample was prepared using a focused ion beam and imaged using the STEM (200 kV). Right: Analyses of the elemental compositions of the dielectric layer, which includes hafnium, oxygen, and aluminum, performed using an energy-dispersive X-ray spectroscopy system. These characterizations show that the thickness of both the Al$_2$O$_3$ and HfO$_2$ layers is ~5 nm.

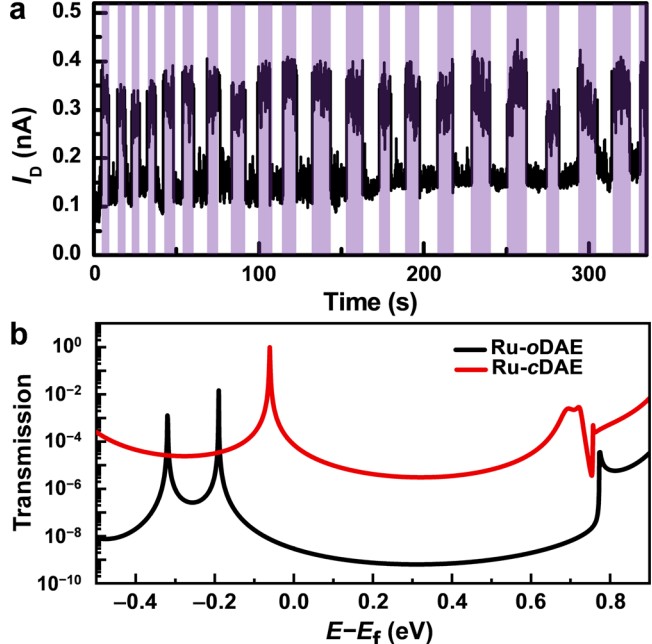

**Fig. 2 Reversible photoswitching of graphene-Ru-DAE-graphene single-molecule junctions. a** Real-time measurement of the current passing through a diarylethene molecule that switches reversibly between its ring-closed and ring-open forms upon exposure to alternate ultraviolet (UV: 380 nm) and visible (Vis: 650 nm) irradiations. Drain voltage $V_D = 300$ mV and gate voltage $V_G = 0$ V. The region with the purple background is under UV irradiation. **b** Transmission spectra of graphene-Ru-DAE-graphene single-molecule junctions with ring-open (dark) and ring-closed (red) isomers. Reprinted with permission from ref. [36]. Copyright 2021 American Chemical Society.

demonstrated below. By applying a dash-line lithographic method, we built nanogapped graphene point contacts and then integrated individual Ru-DAE molecules through covalent amide bonds into electrical nanocircuits. Details of device fabrication and molecular connection are provided in the Supplementary Information (Supplementary Figs. 4, 7, and 8).

DAE is known to be able to isomerize between ring-open and ring-closed forms (Fig. 1a) under light stimuli of different wavelengths[10,25]. Unlike the ring-closed form, which has delocalized π electrons throughout its molecular backbone, the ring-open form demonstrates poor conjugation because the π-electron delocalization is interrupted (the open form is not planar at all) and is restricted to the individual halves of the molecule (Supplementary Fig. 9). Therefore, the Ru-DAE-reconnected single-molecule junctions showed reversible conductance switching behavior under sequential ultraviolet and visible light irradiations in vacuum (Fig. 2a and Supplementary Fig. 18). This photoswitching phenomenon was further verified in solution on a model compound featuring the electrode connection (Supplementary Figs. 3 and 10) and can be well explained based on the results of theoretical transmission spectra. As shown in Fig. 2b, the perturbed highest occupied molecular orbital (*p*-HOMO) of Ru-*c*DAE is closer to the graphene Fermi level than that of Ru-*o*DAE, as expected with the conjugation differences between these two isomers. In addition, the transmission coefficient of the Ru-DAE ring-closed form (Ru-*c*DAE) throughout the energy scale considered here is much higher than that of the Ru-DAE ring-open form (Ru-*o*DAE).

In addition to the optical switching, single-molecule FETs have the high priority for integrated circuits. An important factor that must be considered in the fabrication of high-performance single-molecule FETs is the molecular kernel. The essential condition for realization of a high on/off ratio, which has been a long-sought goal in this field,

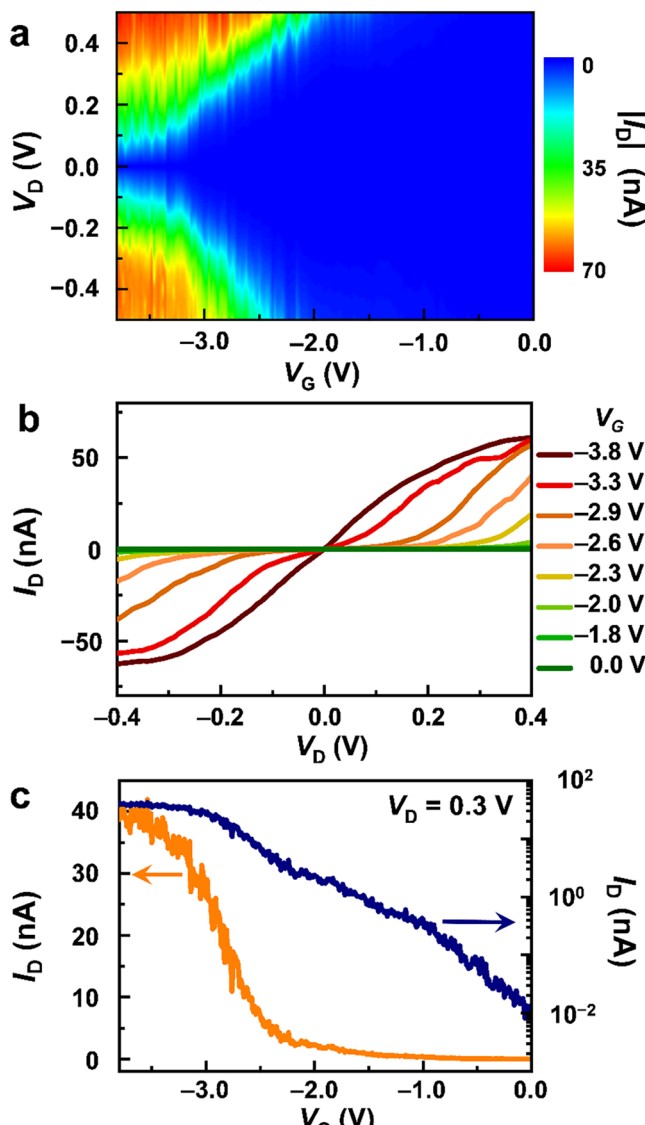

**Fig. 3 Gate-controllable charge transport in Ru-*o*DAE single-molecule transistors. a** Two-dimensional visualization of $I_D$ *vs.* $V_G$ and $V_D$. **b** Representative $I_D$–$V_D$ curves for different values of $V_G$. **c** Transfer characteristics for the Ru-*o*DAE single-molecule FET at $V_D = 0.3$ V.

involves suppression of the off current to the smallest possible level. As reported previously, two novel strategies have been used to realize high on/off ratios: (1) use of the anti-resonance that results from destructive quantum interference[14,15]; and (2) blockading the current in a judiciously designed molecular structure with a small core size and weak electronic molecule-electrode coupling[26]. Comparison of the transmission characteristics of Ru-DAE in the ring-closed and ring-open forms indicates that the current blockade can be realized in Ru-*o*DAE molecular junctions with its lowest transmission at much smaller coefficient, which implies the potential of building single-molecule FETs with ultralow power consumption at the nanometer level because tunneling electrons dominate current at a zero-gate voltage. In addition, because the molecular orbital resonant peak is close to the graphene Fermi level, it can be modulated using the gate voltage to push this resonant peak into the bias window, thus enabling realization of the maximal on-state current at a relatively low gate voltage. Therefore, we should be able to realize the high on/off ratio required by transforming the off-resonant to resonant conducting mechanism via gate control.

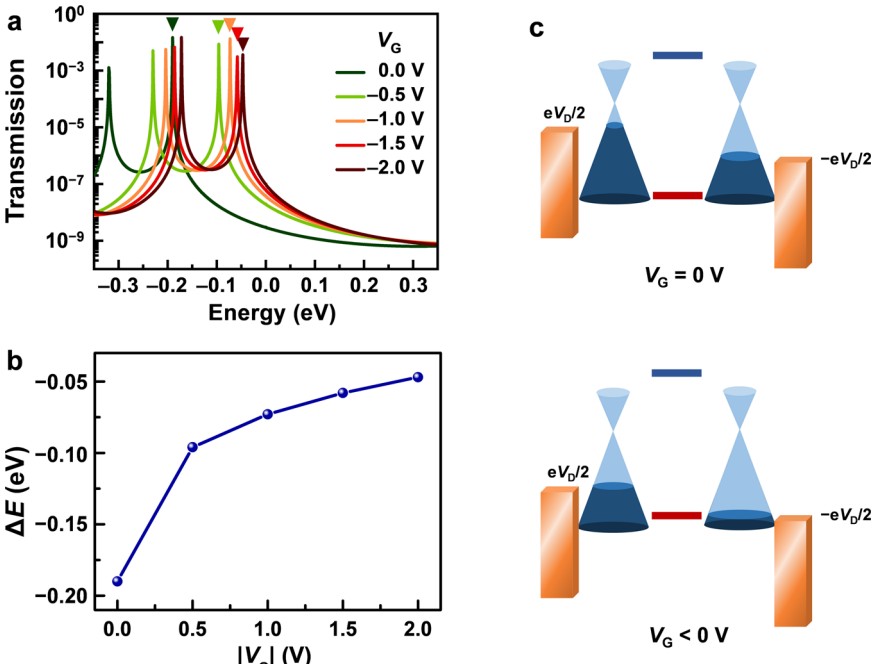

**Fig. 4 Working mechanism of the Ru-oDAE single-molecule FETs. a** Gate-dependent zero-bias transmission spectra at $-2.0\,V \le V_G \le 0\,V$ with steps of 0.5 V. The downward triangles mark the $p$-HOMO for each case. **b** Energy gaps between the $p$-HOMO and the graphene Fermi level at various gate voltages. **c** Schematic energetic diagram showing the alignments between the molecular orbitals (red and blue lines) and the density of graphene electrodes in Ru-oDAE single-molecule transistors under application of different gate voltages.

Based on the prediction described above, we focused on Ru-oDAE for the single-molecule FET study. As Fig. 3 shows, the Ru-oDAE single-molecule transistors presented significant gate-dependent behaviors. In particular, we measured the current-voltage ($I_D-V_D$) curves at negative gate voltages that varied from 0 to $-3.8$ V at intervals of 0.02 V and then compiled these gate-dependent $I_D-V_D$ curves into a current ($|I_D|$) mapping as a function of both $V_D$ and $V_G$, as shown in Fig. 3a. Representative $I_D-V_D$ curves obtained under application of various gate voltages are shown in Fig. 3b and represent a basic feature of a typical FET. The current values at $V_D = 0.3$ V under the different gate voltages were extracted and plotted on both linear and semi-log scales to form the transfer curves shown in Fig. 3c. Remarkably, the current can be modulated by more than three orders of magnitude, while the gate leakage current is negligible (<10 pA; see Supplementary Fig. 11). Both the negligible gate and junction leakages at a zero-gate voltage (~10 pA; see Fig. 3c) provide the opportunity of creating an ultralow power-consumption device as the size of charge transport channel scales down to ~3.1 nm. It should be noted that the extreme difficulty of controlling realistic atomic-level solid-gate interface in experiments will inevitably introduce the quantitative difference of on/off ratios and other FET parameters from one device to another. Nevertheless, the FET phenomenon could be generally obtained by our molecular junctions and appeared in 9 working devices as illustrated in Fig. 3, Supplementary Figs. 12, 13, 19, and 20.

In general, when a negative gate voltage is applied, $p$-HOMO (the dominant conducting molecular orbital) of Ru-oDAE shifts up, thus pushing $p$-HOMO closer to the graphene Fermi level and promoting the conductance. Such gate-induced intrinsic mechanism is well verified by our further theoretical quantum transport calculations on the two-probe Ru-oDAE systems under different gate voltages. Figure 4a presents the variations in the zero-bias transmission spectra under the different gate voltages: As the gate voltage becomes more negative, the transmission spectra move upward in energy overall relative to the graphene

Fermi level (set at zero). Figure 4b demonstrates the non-linear relationship between the gate voltage and the energy gap, which clearly reveals the significant FET behavior at low gate voltage. This is consistent with the behavior shown in the $I-V$ curves (Fig. 3b, c, Supplementary Figs. 12, 15–17, and Supplementary Table 1). In addition, we also analyzed the experimental data with the single-level model[27], and found that the peak bias voltage $V_C$ changes in a similar non-linear behavior. The peak bias $V_C$ disappears when the gate voltage reaches $-3.5$ V (Supplementary Figs. 15–17), which demonstrates that the position of $p$-HOMO gradually approaches the Fermi level of graphene and finally matches the graphene Fermi level as the gate voltage is around $-3.5$ V. As a control group, we also performed similar theoretical calculations for Ru-cDAE (Supplementary Fig. 14), which demonstrated an essentially different FET behavior from that of Ru-oDAE: the transmission spectra move almost linearly with the increasing of gate voltages. From this perspective, it is reasonable to conclude that the intrinsic properties of Ru-oDAE contribute to the performances of these single-molecule FETs with an unprecedentedly high on/off ratios. Moreover, it should be noted that in our experimental setup, the unique electronic properties of the graphene electrodes (Fig. 4c) could also promote the FET behavior because more conducting modes in the graphene electrodes are involved when the $V_G$ is loaded. Note that similar phenomena have been reported in other systems in several previous literatures[28,29].

In summary, the present work represents the realization of a rare high-performance FET behavior achieved at the single-molecule level within a solid-state configuration[22]. The successful integration of both photoswitching and a FET into a single device proves that single-molecule electronics can provide an important solution to enable realization of two basic active electronic elements through a bottom-up approach that goes beyond Moore's law. Flexible molecular and dielectric engineering is the key to the successful realization of these elements and offers a wealth of possibilities for construction of high-performance multifunctional

molecular integrated nanocircuits, thus paving the way to move forward from laboratory-based procedures to industrial-scale processing for real applications.

## Methods

**Molecular synthesis.** Details about the synthetic procedure are provided in the Supplementary Information.

**Device fabrication and molecular connection.** Gate electrode arrays (8 nm Cr/ 60 nm Au) were patterned by photolithography and thermal evaporation on the silicon wafer with a 300 nm silicon oxide layer. Then, Al was deposited on top of Au/Cr electrodes to be connected. After that, the surface of Al was oxidized to $Al_2O_3$ under air, following which $HfO_2$ was deposited by a sol-gel method[30]. At this stage, the back-gate electrode and dielectric layer were fabricated. Then, high-quality single-layer graphene grown through a chemical vapor deposition process was transferred on the surface of $HfO_2$. Source/drain metal electrode arrays (8 nm Cr/60 nm Au) were patterned by photolithography and thermal evaporation. The devices with carboxylic acid-terminated nanogapped graphene point contacts were fabricated by a dash-line lithography method which has been reported previously[31]. Individual diarylethene molecules were connected to graphene point contacts by a dehydration reaction. In brief, diarylethene molecules were dissolved in anhydrous pyridine with the concentration about $10^{-4}$ M. Then, the freshly-cut graphene devices and 1-ethyl-3-(3-dimethylaminopropyl) carbodiimide hydrochloride (EDCI), a well-known carbodiimide dehydrating/activating agent, were added to the solution for connection. After two days in dark and argon atmosphere, the devices were taken out from the solution, cleaned sequentially by ultrapure water and acetone, and dried by nitrogen gas.

**Device measurement.** Device characterizations were carried out under vacuum at 80 K by utilizing a Keysight B1500A semiconductor characterization system and a ST-500-Probe station (Janis Research Company) with a liquid nitrogen cooling system. For $I-V$ measurements, the scanning interval of $V_D$ is 10 mV/step. For $I-t$ measurements, the integration time is 50 ms/step.

**Theoretical calculation.** The structural relaxations of the two-probe transport junctions were performed by the density functional theory (DFT) based on the projector augmented wave method, as implemented in the Vienna Ab initio simulation package (VASP)[32]. The exchange–correlation was treated at the Perdew–Burke–Ernzerhof generalized gradient approximation (PBE-GGA) level[33]. The plane wave basis was set to have a kinetic energy cutoff of 350 eV. The relaxation was deemed to be complete when the residual force on each atom was less than 0.05 eV/Å. The charge transport properties of the two-probe systems with different gate voltages were obtained by using DFT within the nonequilibrium Green's function (NEGF) formalism, as implemented in the quantum transport package Nanodcal[34,35]. The double–zeta polarized atomic orbital basis set was used for all atoms in the NEGF-DFT calculations, and the exchange–correlation was treated at the PBE-GGA level. The cutoff energy for the real–space grid was set at 1360 eV. The NEGF-DFT self–consistent calculations were deemed converged when every element of the Hamiltonian matrix and the density matrix were converged to less than $10^{-4}$ a.u. The transmission spectra were calculated using the Green's function methods. The gate voltage was considered by self-consistently solving Poisson's equation with specific boundary conditions when performing the NEGF-DFT procedure.

## Data availability

All data needed to evaluate the conclusions of this study are available in the main text or Supplementary Materials. The data that support the findings of this study are available from the corresponding authors upon reasonable request.

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

## Acknowledgements

We thank Yunyi Fu and Zhongzheng Tian for the useful discussion. We acknowledge primary financial support from the National Key R&D Program of China (2017YFA0204901, 2021YFA1200101, 2016YFA0300902, and 2019YFA0308500), the National Natural Science Foundation of China (22150013, 21727806, 21933001,

51991344, 91850120, and 11934003), the Natural Science Foundation of Beijing (Z181100004418003 and 2222009), "Strategic Priority Research Program (B)" of Chinese Academy of Sciences (Grant No. XDB330301), the University of Rennes 1, the CNRS and the Agence Nationale de la Recherche (RuOxLux-ANR-12-BS07-0010-01), the Natural Sciences and Engineering Research Council (NSERC) of Canada, and the Fonds de recherche du Quebec-Nature et technologies (FRQNT) of the Province of Quebec (H.G.), Frontiers Science Center for New Organic Matter at Nankai University (63181206) and the Tencent Foundation through the XPLORER PRIZE. The authors also thank the High-Performance Computing Centre of McGill University, CalcuQuebec, and Compute Canada for computation facilities.

## Author contributions

X.G., S.M., and S.R. conceived and designed the experiments. L.M., N.X., and H.J. fabricated the devices. H.A.S., X.H., P.S., L.N., and S.R. did the molecular synthesis. L.M., N.X., Y.J., Q.Z., Z.Y., M.Z., and L.G. performed the device measurements. C.H. and H.G. performed the theoretical calculations. X.G., S.M., H.G., S.R., L.M., N.X., and C.H. analyzed the data and wrote the paper. All the authors discussed the results and commented on the manuscript.

## Competing interests

The authors declare no competing interests.
