## [Peer Review File · Nature Communications]

Reviewers' Comments:

Reviewer #1:

Remarks to the Author:

The manuscript presents an interesting study of a ruthenium-diarylethene complex contacted via graphene electrodes. The authors present data showing electrostatic gate-controlled conductance properties of their molecular junctions (Fig. 3, SI Fig.12) and show a conductance modulation effect for one junction (Fig. 2). It is unclear whether the optical modulation effect and the electrostatic gating effect were observed on the same device.

In the title, the authors speak about "dual-gated" single-molecule transistors and state in abstract (p.3, l.41) that "Both experimental and theoretical results demonstrate these distinct dual-gated behaviors consistently...". I guess that the authors want to point out that one gate is the electrostatic gate and the other the light-modulation effect. The claim of a dual-gating effect is however not supported by the data presented. While the authors show a light-induced conductance modulation for one device at a fixed gate and bias voltage (Fig. 2) and show an electrostatic gating effect on other devices (Fig. 3 and Supp. Fig. 12, showing quite interesting data for such complex devices), it is not clear to me that the data in Fig. 2 are taken on the same device as that of Fig.3. Furthermore, my understanding of a dual gating consists in demonstrating (on the same device) the combined effect of, in this case, the optical and the electrostatic gating and their interplay; and this for several devices to demonstrate reproducibility. As an example, see for instance Katsouris et al., *Sci. Rep.* (2015) who illustrate the interplay between a ferroelectric gate and an electrostatic gate exploited to optimize the on/off ratio in a ferroelectric transistor. As another example, see e.g.: The authors do not demonstrate such a dual-gating capability in their devices here. Control experiments showing e.g. the absence of optical switching prior to molecules deposition are also not shown.

The authors state (p.5, l.94ff) that their experimental approach provides "a reliable (i.e., stable and reproducible) exploration of the device function from the perspective of the intrinsic properties of the connected molecule." While they support this statement by citing Ref. 17 (their own work), I would find it not only appropriate but also necessary to quantify and benchmark their statement with respect to other works and by providing yield, stability and reproducibility statistics for their current study. The stability of graphene electrodes has been demonstrated in the past (see e.g. Prins et al., *Nano Lett.*, 2011, 11, 4607) and the stability of molecular junctions based on graphene electrodes as well (see e.g.: Sun et al., *ChemPhysChem*, 2018, 19, 2258). In Supp. Fig.12, the authors show 4 additional molecular devices and demonstrate some reproducibility indeed. However, from a quick comparison between all devices, one can see that the onset of current takes place at substantially different gate voltages (from a coarse reading, between $-0.5V$ and $-3.0V$) and that the maximum current varies substantially (by more than one order of magnitude) between different devices at similar gate and voltage bias. This comparison is actually not easy as the scales for the gate voltage (x-axis) and the current (color coding) are not the same, the scale for the current varying for instance between $0-3nA$ and $0-250nA$. I do appreciate that these are quite delicate devices to fabricate and delicate measurements but this shows that a clear, quantitative statement of what is considered acceptable to be qualified of reproducible or reliable is necessary and should be substantiated within a more statistical approach.

Two additional comments to the abstract and conclusion if I may. The sentence (p.3, l.42) "...which revolutionises the current technology for creation of practical ultraminiaturised functional electrical circuits beyond Moore's law." is overstated for me. For such a sentence to be justified, the authors would need to demonstrate the integration of several molecular devices in a circuit with a proven electronic transfer function observed, and at least propose a credible technical upscaling pathway providing a reasonable device yield. Please also note that large on/off ratios have been observed in gated atomic-scale junctions since many years, see for instance Martin et al., *Nano Lett.* 2009, 9, 8, 2940. The opening statement in the conclusion (p.11, l.212) that "...the present work represents the first demonstration of a high-performance FET behavior achieved at the single-molecule level within a solid-state configuration." seems therefore also excessive.

While I find the study quite interesting and the concept attractive, the discrepancy between the major claims made in the manuscript and the data shown make me conclude that I cannot support

publication of this work in Nature Communications. At this point, it seems that quite some additional data is needed to justify the claims in the paper, starting with that of dual-gating.

Reviewer #2:

Remarks to the Author:

In this elegant paper the authors present a methodology to perform a single (?) molecular transistor which can be also be switchable by light. The work is nicely presented and the novelty of the transistor in which the thin graphene layer allows the penetration of the gate voltage should be appreciated.

In this stage, I would certainly recommend publishing the paper after addressing the following points:

1. How the author know that indeed the molecule is attached to the graphene layer, and more importantly, how they can prove that they have a single -molecule as claimed?
2. In the field of single molecular electronics reporting on the statistics is essential. This should include error bars as well as a report on the success rate (how many junctions worked)
3. I would highly recommend making an effort and characterize the nanogap since it is claimed that the single molecules is trapped inside
4. The authors present the transport characteristics of the transistor, but not it's characteristic under switching? Highly recommend including this feature

Reviewer #3:

Remarks to the Author:

The manuscript "Dual-gated single-molecule field-effect transistors beyond Moore's law" by Meng et al. describes the placement of a photoswitching molecule between graphene electrodes, and the effects of both optical and electric fields on the transport properties. This is a follow-up paper on the technique that initially showed photoswitching between two electrodes in a similar setup. The fabrication for these devices is difficult in general, and the experimental results are quite clear. The paper itself is somewhat disjointed with a first discussion of photoswitching and then a followed by the gating effect. However, the major issue is the interpretation of the results is ambiguous and confusing. The theoretical description and rationalization of the results need to be re-examined, and likely completely re-written before acceptance anywhere.

The primary issue with the explanation of the experimental observables is that the authors state that the molecular energy levels move up and the graphene fermi energy moves down with the application of a negative gate bias. This is not expected, not clear why it would happen, and contrary to the results in the supplement for a pristine graphene device in which the device becomes more p-type with a negative gate voltage. It is unclear how the authors come to this conclude that the result is the opposite when a molecule is present, and if true should be supported by some experimental evidence. Along these lines, it appears that the transport calculations do not include the graphene electrodes, and if that is the case, then there needs to be both a theoretical and experimental evidence to come to this conclusion. In addition, I'd recommend the authors consider alternative potential working mechanisms for the effect including the partial screening of the gate field in the graphene electrodes (i.e. differential gating of the electrodes and the molecule).

In addition, fitting the I-VSD data with a single-level model would significantly add to any interpretation as it would tell how the alignment between the molecular orbital and the graphene electrodes change with gating.

Minor issues:

On line 89 the authors state "atomic stiffness" but I believe they mean to say "atomic thickness".

The fabrication is difficult and time consuming. The methods section for device fabrication and molecular junction formation should be greatly expanded.

Listed below are the major changes in the new version of the manuscript.

1. We replaced Fig. 2a with photoswitching over 20 cycles in the revised manuscript.
Please see Fig. 2a in the revised Manuscript.
2. One sentence in abstract was modified.
Please see Lines 34–36, Page 3 in the revised Manuscript: “which revolutionises the current technology for creation of practical ultraminiaturised functional electrical circuits beyond Moore’s law” was changed into “which helps to develop the different technology for creation of practical ultraminiaturised functional electrical circuits beyond Moore’s law.”
3. One sentence in summary was modified.
Please see Lines 192–193, Page 10 in the revised Manuscript, “the present work represents the first demonstration of a high-performance FET behavior achieved at the single-molecule level within a solid-state configuration” was modified as “the present work represents the realisation of a rare high-performance FET behavior achieved at the single-molecule level within a solid-state configuration”.
4. We modified the sentence about the relation between *p*-HOMO and the graphene Fermi level and added the comparison between experimental data and simulated results.
Please see Lines 163–165 and Lines 174–179, Page 9 in the revised Manuscript.
5. The detailed information of device fabrication was added in the revised manuscript.
Please see Lines 401–415, Pages 19–20 in the revised Manuscript.
6. Further description about single-molecule connection was added in the revised manuscript.
Please see Lines 430–432 in Page 20 in the revised Manuscript.
7. We added 6 more references in the revised manuscript.
Please see Refs. 18, 19, 22, 27, 33 and 36 in the revised Manuscript.
8. We added the description of reproducible working devices in the main text, updated Extended Data Fig. 12, and added Extended Data Figs. 13, 15, 16, 17, 19 and 20 in the revised Supporting Information.
Please see Lines 157–158, Page 8 in the main text and Extended Data Figs. 12, 13, 15, 16, 17, 19 and 20 in the revised Supporting Information.
9. The original Fig. 2a in the previous Manuscript was moved to the Supporting Information as Extended Data Fig. 18a and a control experiment was added Extended Data Fig. 18b.
Please see Extended Data Fig. 18 in the revised Supporting Information.
10. We added Extended Data Table 1 in the revised Supporting Information.
Please see Extended Data Table 1 in the revised Supporting Information.

Reviewer #1 (Remarks to the Author):

The manuscript presents an interesting study of a ruthenium-diarylethene complex contacted via graphene electrodes. The authors present data showing electrostatic gate-controlled conductance properties of their molecular junctions (Fig. 3, SI Fig. 12) and show a conductance modulation effect for one junction (Fig. 2). It is unclear whether the optical modulation effect and the electrostatic gating effect were observed on the same device.

General reply: Thank the referee very much for his/her high evaluation and constructive suggestions! After we carefully revised the article fully according to your suggestions as detailed below, the manuscript has been significantly strengthened.

1. In the title, the authors speak about “dual-gated” single-molecule transistors and state in abstract (p.3, 1.41) that “Both experimental and theoretical results demonstrate these distinct dual-gated behaviors consistently...”. I guess that the authors want to point out that one gate is the electrostatic gate and the other the light-modulation effect. The claim of a dual-gating effect is however not supported by the data presented. While the authors show a light-induced conductance modulation for one device at a fixed gate and bias voltage (Fig. 2) and show an electrostatic gating effect on other devices (Fig. 3 and Supp. Fig. 12, showing quite interesting data for such complex devices), it is not clear to me that the data in Fig. 2 are taken on the same device as that of Fig. 3. Furthermore, my understanding of a dual gating consists in demonstrating (on the same device) the combined effect of, in this case, the optical and the electrostatic gating and their interplay; and this for several devices to demonstrate reproducibility. As an example, see for instance Katsouris et al., Sci. Rep. (2015) who illustrate the interplay between a ferroelectric gate and an electrostatic gate exploited to optimize the on/off ratio in a ferroelectric transistor. As another example, see e.g.: The authors do not demonstrate such a dual-gating capability in their devices here. Control experiments showing e.g. the absence of optical switching prior to molecules deposition are also not shown.

Our reply: Thanks for the nice comment. Sorry for the misunderstanding. The device, which has the photoswitching effect in Extended Data Fig. 18a, also shows the gating effect as presented in Extended Data Fig. 12c. The big challenge we usually meet is that the junctions break down due to the unexpected electrostatic field from the environment before achieving the gating effect under alternate UV and visible illumination.

Fully according to this reviewer’s suggestion, to prove this dual gating effect, with great efforts, we carried out additional huge experiments and provided corresponding data as shown in Fig. 2a (Fig. R1-1) and Extended Data Fig. 19 (Fig. R1-2). The photoswitching and gating effect for the ring-closed and ring-open forms were obtained on the same device. The conductance of the ring-closed form is larger than the ring-open form, while the field effect performance of the ring-open form is

better than that of the ring-closed form, which is consistent with the theoretical simulations as shown in Fig. 4 and Extended Data Fig. 14.

In addition, control experiments showing the absence of optical switching prior to molecules deposition are provided accordingly as Extended Data Fig. 18b (Fig. R1-3).

Our revision: We added the experimental data of photoswitching in Fig. 2a (Fig. R1-1) in the revised manuscript, the corresponding gate effect data in Extended Data Fig. 19 (Fig. R1-2) and the result from control experiments in Extended Data Fig. 18b (Fig. R1-3) in the revised Supporting Information.

Fig. R1-1 (Fig. 2a) | Reversible photoswitching of graphene-Ru-DAE-graphene single-molecule junctions. Real-time measurement of the current passing through a diarylethene molecule that switches reversibly between its ring-closed and ring-open forms upon exposure to alternate ultraviolet (UV: 380 nm) and visible (Vis: 650 nm) irradiations. Drain voltage $V_D = 300$ mV and gate voltage $V_G = 0$ V. The region with the purple background is under UV irradiation.

Fig. R1-2 (Extended Data Fig. 19) | The gating effect of ring-open and ring-closed states for the same device in Fig. 2a. (a, c and e) are under visible irradiation and (b, d and f) are after UV irradiation. (a and b) Two-dimensional visualization of I_D vs. V_G and V_D . V_G step is 500 mV. (c and d) Representative I_D vs. V_D curves for different values of V_G . (e and f) Corresponding transfer characteristics at $V_D = 300$ mV.

Fig. R1-3 (Extended Data Fig. 18b) | Control experiments showing the absence of optical switching prior to molecules deposition. $V_D = 100$ mV and $V_G = 0$ V.

2. The authors state (p.5, 1.94ff) that their experimental approach provides “a reliable (i.e., stable and reproducible) exploration of the device function from the perspective of the intrinsic properties of the connected molecule.” While they support this statement by citing Ref. 17 (their own work), I would find it not only appropriate but also necessary to quantify and benchmark their statement with respect to other works and by providing yield, stability and reproducibility statistics for their current study. The stability of graphene electrodes has been demonstrated in the past (see e.g. Prins et al., Nano Lett., 2011, 11, 4607) and the stability of molecular junctions based on graphene electrodes as well (see e.g.: Sun et al., ChemPhysChem, 2018, 19, 2258). In Supp. Fig. 12, the authors show 4 additional molecular devices and demonstrate some reproducibility indeed. However, from a quick comparison between all devices, one can see that the onset of current takes place at substantially different gate voltages (from a coarse reading, between -0.5V and -3.0V) and that the maximum current varies substantially (by more than one order of magnitude) between different devices at similar gate and voltage bias. This comparison is actually not easy as the scales for the gate voltage (x-axis) and the current (color coding) are not the same, the scale for the current varying for instance between $0\text{-}3\text{nA}$ and $0\text{-}250\text{nA}$. I do appreciate that these are quite delicate devices to fabricate and delicate measurements but this shows that a clear, quantitative statement of what is considered acceptable to be qualified of reproducible or reliable is necessary and should be substantiated within a more statistical approach.

Our reply: Thanks for the good comments. We really appreciate the reviewer’s understanding of this work, including the difficulty of device fabrication and delicate measurements. On the basis of our previous works, our single-molecule devices are stable and can endure external stimuli and long-term measurements because single molecules are covalently connected with graphene electrodes through robust amide linkages. To further demonstrate the device stability, we have added two works on other graphene-based single-molecule junctions as Refs. 18 and 19 as this reviewer mentioned.

We succeeded in integrating the gate electrode and the ultrathin dielectric layer into graphene-based single-molecule junctions, which is really challenging. However, the roughness of gate dielectric under graphene and the junction is larger than silicon substrates. The quality and thickness of the dielectric layer vary among different devices. In general, these tiny changes of each parameter could lead to the variation in the device performance. Therefore, the maximum gate voltage was chosen as the leakage current starts to increase sharply, which results in the different ranges of the gate voltage.

The scale for the current varying between $0\text{-}3\text{ nA}$ and $0\text{-}250\text{ nA}$ mainly originates from the heterogeneity of single-molecule devices. Although the currents vary in the range of around 1 order of magnitude as the gate voltage varies within 1 V , the gate effect is similar for all the devices as shown in Figures 3 and S12 as well as the figure below (Fig. R1-4) obtained from additional huge works ($\sim 5\%$

yield, totally reproducible 9 working devices). Conductance variations is a general phenomenon among nano/molecular devices. For example, in the work based on MCBJ (*Sci. Adv.* **2017**, 3, eaao2615), the range is over 1 order of magnitude, too. Realization of atomic level precision in the cutting procedure and precise control of the molecular conformation on the substrate within the graphene gaps and the contact configuration are the challenges for future studies to overcome, which we are developing currently.

Our revision: Two new references were added as Refs. 18 and 19 in the revised Main Manuscript. Statistic distribution of on/off ratios among different devices in the ring-open state at $V_D = 300$ mV was added as Extended Data Fig. 20 (Fig. R1-4) in the revised Supporting Information.

Ref. 18: Prins, F. *et al.* Room-Temperature Gating of Molecular Junctions Using Few-Layer Graphene Nanogap Electrodes. *Nano Lett.* **11**, 4607–4611 (2011).

Ref. 19: Sun, H., Jiang, Z., Xin, N., Guo, X., Hou, S. & Liao, J. Efficient Fabrication of Stable Graphene-Molecule-Graphene Single-Molecule Junctions at Room Temperature. *ChemPhysChem* **19**, 1–9 (2018).

Fig. R1-4 (Extended Data Fig. 20) | Statistic distribution of on/off ratios among different devices in the ring-open state at $V_D = 300$ mV.

- Two additional comments to the abstract and conclusion if I may. The sentence (p.3, l.42) "...which revolutionises the current technology for creation of practical ultraminiaturised functional electrical circuits beyond Moore's law." is overstated for me. For such a sentence to be justified, the authors would need to demonstrate the integration of several molecular devices in a circuit with a proven electronic transfer function observed, and at least propose a credible technical upscaling pathway providing a reasonable device yield. Please also note that large on/off ratios have been observed in gated atomic-scale junctions since many years, see for instance Martin et al., *Nano Lett.* 2009, 9, 8, 2940. The opening statement in the

conclusion (p.11, l.212) that "...the present work represents the first demonstration of a high-performance FET behavior achieved at the single-molecule level within a solid-state configuration." seems therefore also excessive.

Our reply: Thanks for the invaluable suggestions. Fully according to the suggestion of this reviewer, "which revolutionises the current technology for creation of practical ultraminiaturised functional electrical circuits beyond Moore's law" in Line 42, Page 3 was revised as "which helps to develop the different technology for creation of practical ultraminiaturised functional electrical circuits beyond Moore's law." in the revised manuscript and "the present work represents the first demonstration of a high-performance FET behavior achieved at the single-molecule level within a solid-state configuration" in Line 190, Page 10 was modified as "the present work represents the realisation of a rare high-performance FET behavior achieved at the single-molecule level within a solid-state configuration". We also added a reference as Ref. 22 as the reviewer mentioned.

Our revision: "which revolutionises the current technology for creation of practical ultraminiaturised functional electrical circuits beyond Moore's law" in Line 42, Page 3 was revised as "which helps to develop the different technology for creation of practical ultraminiaturised functional electrical circuits beyond Moore's law." in the revised manuscript (Lines 34–36, Page 3). "The present work represents the first demonstration of a high-performance FET behavior achieved at the single-molecule level within a solid-state configuration" in Line 192, Page 10 was modified as "the present work represents the realisation of a rare high-performance FET behavior achieved at the single-molecule level within a solid-state configuration." in the revised manuscript (Lines 192–193, Page 10).

One additional reference was added as Ref. 22.

Ref. 22: Martin, C. A., Smit, R. H. M., van der Zant, H. S. J. & van Ruitenbeek, J. M. A nanoelectromechanical single-atom switch. *Nano Lett.*, **9**, 2940–2945 (2009).

While I find the study quite interesting and the concept attractive, the discrepancy between the major claims made in the manuscript and the data shown make me conclude that I cannot support publication of this work in Nature Communications. At this point, it seems that quite some additional data is needed to justify the claims in the paper, starting with that of dual-gating.

Thank this reviewer very much for the useful suggestions! We have carried out the huge works and added more experimental data to verify the dual gating effect on the same device.

Reviewer #2 (Remarks to the Author):

In this elegant paper the authors present a methodology to perform a single (?) molecular transistor which can be also be switchable by light. The work is nicely presented and the novelty of the transistor in which the thin graphene layer allows the penetration of the gate voltage should be appreciated.

General reply: Thank the referee very much for his/her high appraisal and kind recommendation. Listed below are the point-by-point responses for each comment. After revisions, the manuscript has been significantly strengthened.

In this stage, I would certainly recommend publishing the paper after addressing the following points:

1. How the author know that indeed the molecule us attached to the graphene layer, and more importantly, how they can prove that they have a single -molecule as claimed?

Our reply: Thanks for the nice comments. The success of molecular connection is generally proved by the comparison of the conductance difference before RIE cutting and after further molecular connection. As shown in Extended Data Fig. 8 (also see Figure R2-1), the junction conductance at 0.5 V is around 4 orders of magnitude higher after a Ru-DAE molecule is connected between graphene electrodes than that after RIE cutting, which is basically the noise of the instrument. In addition, the optical switch shown in Fig. 2a and Extended Data Fig. 18 is another solid evidence of a single molecule of Ru-DAE on the junction.

Fig. R2-1 (Extended Data Fig. 8): I - V curves of a device at different stages with semi-log scales: nanogapped graphene (grey solid line) and Ru-DAE-connected graphene (blue solid line). The gate voltage is 0 V.

To prove that each device has a single molecule, as demonstrated in the Supporting Information, the number of junctions that contribute to charge transport is carried out by calculating the probability of the connected devices with n -rejoined junctions (G_n) with the binomial distribution and the optimized connection yields:

$$G_n = \frac{m!}{n!(m-n)!} p^n (1-p)^{m-n} \quad n = 0, 1, 2 \dots, m$$

Supplementary Equation 1

where m is the number of graphene point contact pairs (210 in the current case) and p is the probability of successful connection for a random junction. Therefore, the possibility of connected junction γ_c can be attained:

$$\gamma_c = 1 - G_0 = 1 - \frac{m!}{0!(m-0)!} p^0 (1-p)^m = 1 - (1-p)^m$$

Supplementary Equation 2

where G_0 is the probability of devices without any connected junctions. In our experiments, ~5% junctions show successful connection of molecules. Then, the ratio of single-junction devices to the overall reconnected devices is ~97.5%. These results from calculations suggest that, in most cases, charge transport in these devices arises mainly in a single-molecule junction. Another direct evidence of the successful connection of a single molecule in our devices can be found in our other recent works (*Sci. Adv.* **2021**, *7*, eabf0689; *Nat. Nanotechnol.* **2021**, DOI: <https://doi.org/10.1038/s41565-021-00959-4>), where we used a self-built superhigh resolution fluorescence microscopy to achieve single-molecule fluorescent images (Please see Fig. R2-2).

Fig. R2-2: Fluorescent super-resolution imaging of the single-molecule connection (Figure 1B from *Sci. Adv.* **2021**, *7*, eabf0689).

Our revision: “Further experimental evidence for the successful formation of GMG-SMJs can be found in a recent work³⁶, where we used a self-built photoelectrical integrated characterization system to demonstrate the single-molecule connection” was added on Lines 430–432, Page 20 in the revised manuscript.

Ref. 36: Yang, C., *et al.* Electric field-catalyzed single-molecule Diels-Alder reaction dynamics. *Sci. Adv.* **7**, eabf0689 (2021).

- In the field of single molecular electronics reporting on the statistics is essential. This should include error bars as well as a report on the success rate (how many junctions worked)

Our reply: Thanks for the useful suggestion. The working junctions are listed in the main manuscript and Supporting Information as well as the transfer curves with logarithm coordinate. There are total 9 working junctions as show in Fig. 3, Extended Data Figs. 12, 19 and 20 from tremendous works.

Our revision: Two new figures were added as Extended Data Figs. 19 and 20 (Fig. R1-2 and Fig. R1-4) in the revised Supporting Information.

Fig. R1-2 (Extended Data Fig. 19) | The gating effect of ring-open and ring-closed states for the same device in Fig. 2a. (a, c and e) are under visible irradiation and (b, d and f) are after UV irradiation. (a and b) Two-dimensional visualization of I_D vs. V_G and V_D . V_G step is 500 mV. (c and d) Representative I_D vs. V_D curves for different values of V_G . (e and f) Corresponding transfer characteristics at $V_D = 300$ mV.

Fig. R1-4 (Extended Data Fig. 20) | Statistic distribution of on/off ratios among different devices in the ring-open state at $V_D = 300$ mV.

- I would highly recommend making an effort and characterize the nanogap since it is claimed that the single molecules is trapped inside.

Our reply: Thanks for the nice suggestion. The existence of a single molecule has been proved by the response to Question 1 as follows: 1, The junction conductance recovers after the functional molecule is connected between graphene electrodes, as shown Extended Data Fig. 8 and Figure R1-2. 2, The photoswitching effect in Fig. 2A and Extended Data Fig. 18 provide another solid evidence of a single molecule of Ru-DAE on the junction. 3, The binomial distribution calculation demonstrates that charge transport in these devices arises mainly in a single-molecule junction. 4, Further experimental evidence for the successful formation of Ru-DAE single-molecule junctions can be found in a recent work (*Sci. Adv.* **2021**, 7, eabf0689), where we used a self-built photoelectrical integrated characterization system to demonstrate the single-molecule connection. In addition, as shown in Figure 1a, the AFM image clearly shows the gap and the device architecture at the nanometer level. More AFM images can be found in a previous work (*Angew. Chem. Int. Ed.* **2012**, 51, 12228–12232).

Our revision: “Further experimental evidence for the successful formation of GMG-SMJJs can be found in a recent work, where we used a self-built photoelectrical integrated characterization system to demonstrate the single-molecule connection” was added on Lines 430–432, Page 20 in the revised manuscript.

- The authors present the transport characteristics of the transistor, but not its characteristic under switching? Highly recommend including this feature.

Our reply: Thanks for the useful suggestion. Fully according to this suggestion, the gate effect of photoswitching under UV/vis irradiation have been added as revised Fig. 2A in the revised manuscript. The gating effect for both ring-open and ring-

closed states in the same device is also added as Extended Data Fig. 19 in the revised Supporting Information. The characteristics of gating effect on ring-open and ring-closed forms are consistent with the simulation in Fig. 4 and Extended Data Fig. 14, proving that the field effect performance of the open-ring form is better than that of the closed-ring form. In addition, based on the simulation results, the *p*-HOMO of Ru-*c*DAE is closer to the Fermi level of graphene, and the difference between the transmission at the peak of *p*-HOMO and the Fermi level is much smaller than Ru-*o*DAE, which results in lower on/off ratio for Ru-*c*DAE.

Our revision: We added the photoswitching phenomenon in Fig. 2a (Fig. R1-1) as well as the gate effect on both open and close forms in Extended Data Fig. 19 (Fig. R1-2) in the revised manuscript.

Fig. R1-1 (Fig. 2a) | Reversible photoswitching of graphene-Ru-DAE-graphene single-molecule junctions. Real-time measurement of the current passing through a diarylethene molecule that switches reversibly between its ring-closed and ring-open forms upon exposure to alternate ultraviolet (UV: 380 nm) and visible (Vis: 650 nm) irradiations. Drain voltage $V_D = 300$ mV and gate voltage $V_G = 0$ V. The region with the purple background is under UV irradiation.

Fig. R1-2 (Extended Data Fig. 19) | The gating effect of ring-open and ring-closed states for the same device in Fig. 2a. (a, c and e) are under visible irradiation and (b, d and f) are after UV irradiation. (a and b) Two-dimensional visualization of I_D vs. V_G and V_D . V_G step is 500 mV. (c and d) Representative I_D vs. V_D curves for different values of V_G . (e and f) Corresponding transfer characteristics at $V_D = 300$ mV.

Reviewer #3 (Remarks to the Author):

The manuscript “Dual-gated single-molecule field-effect transistors beyond Moore’s law” by Meng et al. describes the placement of a photoswitching molecule between graphene electrodes, and the effects of both optical and electric fields on the transport properties. This is a follow-up paper on the technique that initially showed photoswitching between two electrodes in a similar setup. The fabrication for these devices is difficult in general, and the experimental results are quite clear. The paper itself is somewhat disjointed with a first discussion of photoswitching and then a followed by the gating effect. However, the major issue is the interpretation of the results is ambiguous and confusing. The theoretical description and rationalization of the results need to be re-examined, and likely completely re-written before acceptance anywhere.

General reply: Thank the referee very much for his/her high evaluation and kind recommendation. We have carefully re-examined and re-written the theoretical description and rationalisation of the results as shown below. After revisions, the manuscript has been significantly strengthened.

1. The primary issue with the explanation of the experimental observables is that the authors state that the molecular energy levels move up and the graphene fermi energy moves down with the application of a negative gate bias. This is not expected, not clear why it would happen, and contrary to the results in the supplement for a pristine graphene device in which the device becomes more p-type with a negative gate voltage. It is unclear how the authors come to this conclude that the result is the opposite when a molecule is present, and if true should be supported by some experimental evidence. Along these lines, it appears that the transport calculations do not include the graphene electrodes, and if that is the case, then there needs to be both a theoretical and experimental evidence to come to this conclusion. In addition, I’d recommend the authors consider alternative potential working mechanisms for the effect including the partial screening of the gate field in the graphene electrodes (i.e. differential gating of the electrodes and the molecule).

Our reply: Thanks for the good comments. Sorry for the misunderstanding. The Fermi level of graphene moves down with the application of a negative gate bias. This shift is relative to the graphene Dirac point.

The graphene Fermi level is believed to be the same with the Fermi level of electrodes because graphene is connected with Au electrodes (source and drain) for further test, as shown in Figs. 1b and 4c. The negative gate voltage pulls up the energy level of the orbitals for both the functional molecule and graphene, because the increased additional energy $-\alpha |e| \cdot V_g$ is positive, where the minus is the negative charge of electron and α is the positive coefficient. The negative gate voltage induces their positive additional energy, leading to the shift up of the orbitals for both the molecule and graphene. At the same time, the voltage on source and

drain electrodes is fixed. Therefore, the graphene Dirac point shifts up while the Fermi level is fixed, as shown in Fig. 4c. In addition, the increased carrier density of graphene on the negative gate voltage might make a positive contribution to the conductance. Correspondingly, the main manuscript has been modified as “In general, when a negative gate voltage is applied, *p*-HOMO (the dominant conducting molecular orbital) of Ru-*o*DAE shifts up, thus pushing *p*-HOMO closer to the graphene Fermi level.”. The detailed information and the explanation are clear now. Thanks again for the useful suggestion.

Our revision: The sentence on Lines 163–165, Page 9 was modified as “In general, when a negative gate voltage is applied, *p*-HOMO (the dominant conducting molecular orbital) of Ru-*o*DAE shifts up, thus pushing *p*-HOMO closer to the graphene Fermi level.” in the revised manuscript.

2. In addition, fitting the I-VSD data with a single-level model would significantly add to any interpretation as it would tell how the alignment between the molecular orbital and the graphene electrodes change with gating.

Our reply: Thanks for the constructive suggestion. Fully according to this suggestion, we added the $|V^2/I|$ vs. V curves at different gate voltages as Fig. R3-1 (Extended Data Fig. 15) for the device in Fig. 3 (Ru-*o*DAE) and the comparison of the theoretical I - V curve with the single-level model to the experimental I - V curve at different gate voltage as Fig. R3-2 (Extended Data Fig. 16). The theoretical curves match well with the experimental data. In Fig. R3-1 (Extended Data Fig. 15) and Table R1 (Extended Data Table 1), the peak bias goes from 0.21 V ($V_G = 0.0$ V) and 0.81 V ($V_G = -2.5$ V) to 0.0 V ($V_G = -3.5$ V).

In addition, we added Extended Data Fig. 17 (Fig. R3-3) to show the change of the peak biases in Extended Data Fig. 15 relative to the gate voltage and the corresponding reference as Ref. 27.

Fig. R3-1 (Extended Data Fig. 15) $|V^2/I|$ vs. V curves at different gate voltages for the device in Fig. 3 (Ru-oDAE). The HOMO falls in the Fermi level as the gate voltage goes to -3.5 V.

Fig. R3-2 (Extended Data Fig. 16) | Experimental and theoretical I - V curves for different gate voltages, where theoretically $I = I_R * I_C$ and $V = V_R * V_C$. Based on the law of corresponding states (LCS) for electron tunneling mediated by a single level in molecular junctions, the maxima of $|V^2/I|$ vs. V curves can be used to define “natural” units V_C and I_C for voltage and current, respectively, as shown in Extended data Table 1. The theoretical I - V curves are based on the relationship of dimensionless biases ($V \equiv V_R * V_C$) and currents ($I \equiv I_R * I_C$), where I_R and V_R can be expressed as follows:

$$I_R = \frac{2V_R}{3 - V_R^2}$$

The theoretical gate dependent I - V curves are achieved based on the differences of V_C and I_C in Extended Data Table 1, which vary along with the gate voltage.

Table R3-1 (Extended Data Table 1) | The “natural” units V_C and I_C for different gate voltages.

V_G	$V_{C_negat\ ive}$	$V_{C_posit\ ive}$	V_C	$I_{C_negat\ ive}$	$I_{C_posit\ ive}$	I_C
-0.0 V	-0.191	0.231	0.211	-0.002144	0.00286	0.002502
-0.5 V	-0.115	0.149	0.132	-0.001259	0.002356	0.001808
-1.5 V	-0.110	0.092	0.101	-0.02728	0.02194	0.02461
-2.5 V	-0.088	0.073	0.081	-0.1518	0.1319	0.1419

Fig. R3-3 (Extended Data Fig. 17) | The peak bias in Extended Data Fig. 14 relative to the gate voltage. The energy gap between p -HOMO and the Fermi level of graphene electrodes decreases as the gate voltage becomes more negative.

Our revision: We added the single level model as Extended Data Fig. 15 (Fig. R3-1), the corresponding data as Extended Data Table 1 (Table R3-1), the correlation between gate voltages and peak bias voltage V_C as Extended Data Fig. 16 (Fig. R3-2), and the change of the peak biases as Extended Data Fig. 17 (Fig. R3-3) in the revised Supporting Information. Please see Fig. R3-1, Fig. R3-2, Fig. R3-3 and Table R3-1 as shown above. We also added the corresponding reference as Ref. 27.

Ref. 27: Bâldea, I. *et al.* Uncovering a law of corresponding states for electron tunneling in molecular junctions. *Nanoscale* **7**, 10465–10471 (2015).

We also modified the corresponding description as follows: “In addition, we also analysed the experimental data with the single-level model, and found that the peak bias voltage V_C changes in a slower manner (i.e., it appears to reach saturation) when the gate voltage goes higher. The peak bias V_C disappears when the gate voltage reaches -3.5 V (Extended Data Figs. 15, 16 and 17), which demonstrates that the position of p -HOMO gradually approaches the Fermi level of graphene and finally matches the graphene Fermi level as the gate voltage is around 3.5 V.” Please see Lines 174–179, Page 9 in the revised manuscript.

Minor issues:

3. On line 89 the authors state “atomic stiffness” but I believe they mean to say “atomic thickness”.

Our reply: Thanks for the reviewer’s comment.

Our revision: The “atomic stiffness” in Line 68, Page 4 was modified as “atomic thickness”.

4. The fabrication is difficult and time consuming. The methods section for device fabrication and molecular junction formation should be greatly expanded.

Our reply: Thanks for the reviewer’s suggestion. The detailed information of device fabrication was added in the revised manuscript.”

Our revision: The details of device fabrication can be found in the Methods section as follows (Lines 401–415, Pages 19–20): “**Device fabrication and molecular connection.** The devices were fabricated by photolithography and dash-line electron beam lithography according to the detailed procedure reported previously³². The difference is the fabrication of the gate electrode and the dielectric layer. After the deposition of patterned Au/Cr electrodes as gate, Al was deposited cross Au/Cr electrodes to be connected. Then, the surface of Al was oxidised to Al₂O₃ under air, following which HfO₂ was deposited by a sol-gel method³³. At this stage, the back-gate electrode and dielectric layer were fabricated. Then, graphene was transferred on the surface of HfO₂ for further use as point contact electrodes to connect the molecule. The following procedure has been reported previously³². Individual diarylethene molecules were connected to graphene point contacts by a dehydration reaction. In brief, diarylethene molecules were dissolved in anhydrous pyridine with the concentration about 10⁻⁴ M. Then, the freshly-cut graphene devices and 1-ethyl-3-(3-dimethylaminopropyl) carbodiimide hydrochloride (EDCI), a well-known carbodiimide dehydrating/activating agent, were added to the solution for connection. After two days in dark and argon atmosphere, the devices were taken out from the solution, cleaned sequentially by ultrapure water and acetone, and dried by nitrogen gas.”

Ref. 33: Acton, O., Ting, G., Ma, H., Ka, J. W., Yip, H.-L., Tucker, N. M., Jen, A. K.-Y. π - σ -phosphonic acid organic monolayer/sol-gel hafnium oxide hybrid dielectrics for low-voltage organic transistors. *Adv. Mater.* **20**, 3697–3701 (2008).

Finally, we would like to thank all the referees very much for their patience, time and kind recommendation!

Reviewers' Comments:

Reviewer #1:

Remarks to the Author:

The authors made a substantial effort to address the points raised in the review and better position their work in the current state-of-the-art.
I have no further comments.

Reviewer #2:

None

Reviewer #3:

Remarks to the Author:

I am satisfied with the authors revisions.

Reviewer #1 (Remarks to the Author):

The authors made a substantial effort to address the points raised in the review and better position their work in the current state-of-the-art.

I have no further comments.

Thank this referee very much for his/her approval and kind recommendation!

Reviewer #3 (Remarks to the Author):

I am satisfied with the authors revisions.

Thank this referee very much for his/her approval and kind recommendation!